# A GeoNEX-based high spatiotemporal resolution product of land surface downward shortwave radiation and photosynthetically active radiation

Ruohan Li[1], Dongdong Wang[1], Weile Wang[2], Ramakrishna Nemani[2]

[1]Department of Geographical Sciences, University of Maryland, College Park, MD 20742, USA

[2]NASA Ames Research Center, Mountain View, CA 94043, USA

*Correspondence to*: Dongdong Wang (ddwang@umd.edu)

**Abstract.** Surface downward shortwave radiation (DSR) and photosynthetically active radiation (PAR) play critical roles in the Earth's surface processes. As the main inputs of various ecological, hydrological, carbon, and solar photovoltaic models, increasing requirements for high spatiotemporal resolution DSR and PAR estimation with high accuracy have been observed in recent years. However, few existing products satisfy all of these requirements. This study employed a well-established physical-based look-up table (LUT) approach to the GeoNEX gridded top-of-atmosphere bidirectional reflectance factor data acquired by the Advanced Himawari Imager (AHI) and Advanced Baseline Imager (ABI) sensors. It produced a data product of DSR and PAR over both AHI and ABI coverage at an hourly temporal step with a 1 km spatial resolution. GeoNEX DSR data were validated over 63 stations, and GeoNEX PAR data were validated over 27 stations. The validation showed that the new GeoNEX DSR and PAR products have accuracy higher than other existing products, with root mean square error (RMSE) of hourly GeoNEX DSR achieving 74.3 $W/m^2$ (18.0%), daily DSR estimation achieving 18.0 $W/m^2$ (9.2%), hourly GeoNEX PAR achieving 34.9 $W/m^2$ (19.6%), and daily PAR achieving 9.5 $W/m^2$ (10.5%). The study also demonstrated the application of the high spatiotemporal resolution GeoNEX DSR product in investigating the spatial heterogeneity and temporal variability of surface solar radiation. The data product can be freely accessed through the NASA Advanced Supercomputing Division GeoNEX data portal: https://data.nas.nasa.gov/geonex/geonexdata/GOES16/GEONEX-L2/DSR-PAR/ and https://data.nas.nasa.gov/geonex/geonexdata/HIMAWARI8/GEONEX-L2/DSR-PAR/ (https://doi.org/10.5281/zenodo.7023863, Wang & Li, 2022).

## 1 Introduction

Surface downward shortwave radiation is of great importance to the surface energy balance and hence is the required input of various surface models. Downward shortwave radiation (DSR) is defined as solar radiation received at the Earth's surface within the wavelength range of 300–4000 nm. It is the fundamental driving force of many global ecological, hydrological, and biochemical processes (Huang et al., 2019; Wang et al., 2021; Liang et al., 2019) and provides one of the most promising

renewable energy sources, solar energy. Photosynthetically active radiation (PAR) is the visible component of DSR in the spectral range of 400–700 nm. It also serves as the main input for terrestrial ecosystem modeling, carbon cycle modeling, and yield estimations because of its functionality in photosynthesis (Prince and Goward, 1995; Gu et al., 2002).

The need for a high spatiotemporal DSR product has increased noticeably in recent years. For example, high temporal resolution of solar resource data is required by new power system models, such as the Integrated Grid Modeling System (IGMS) (Palmintier et al., 2017). Additionally, information about the short-term fluctuation of DSR is critical for storage analysis of large grid-connected photovoltaic plants through ramp-rate control (Marcos et al., 2014). High spatial resolution DSR data are prerequisites for producing small-scale solar energy, which has received increased attention in recent years (Jain et al., 2017). Hence, the combination of high spatial and temporal DSR estimations is important for the economic and stable operation of the solar grid (Buster et al., 2021). However, because of the comparatively coarse resolution of existing products, the spatial and diurnal variations in DSR at large scales have not been fully studied. Moreover, as the driving parameters of various land models, DSR and PAR data with high spatial and temporal resolutions are essential for estimating many other surface variables at high spatial and temporal scales, such as land surface temperature (Jia et al., 2022a, 2022b), ground-level ozone mapping (Wei et al., 2022), and evapotranspiration (Huang et al., 2019).

Satellite-based estimation of surface incident shortwave radiation has rapidly developed in recent decades. Polar-orbiting satellite sensors, such as the Moderate Resolution Imaging Spectroradiometer (MODIS), provide one of the most popular data sources because of their extended global coverage and availability of mature high-level atmospheric and surface products (Liang et al., 2006; Zhang et al., 2014; Zhang et al., 2018). Many existing DSR and PAR products, such as the Breathing Earth System Simulator (BESS) (Ryu et al., 2018), Global Land Surface Satellite Downward Shortwave Radiation (GLASS) (Zhang et al., 2019), and MODIS land surface Downward Shortwave Radiation (MCD18) (Wang et al., 2020) were generated based on MODIS observations. With the future retirement of MODIS, studies have started to focus on estimating DSR from the Visible Infrared Imaging Radiometer Suite as well (Li et al., 2022). An advanced very high-resolution radiometer (AVHRR) is also a valuable data source for DSR estimation owing to its long-term data record (Yang et al., 2018). The Clouds, Albedo, and Radiation Edition 2 (CLARA) data products were based on AVHRR (Karlsson et al., 2017). However, the above-mentioned products usually generate daily DSR by interpolating from instantaneous estimation because of the limited overpass counts of polar-orbiting satellites. Because of their limitation to capture diurnal DSR variation, the root mean square error (RMSE) of these products can hardly reach 25 $W/m^2$ for daily DSR (Li et al., 2021).

A lot of efforts have been made to develop the high temporal resolution DSR estimation. The hourly Earth's Radiant Energy System (CERES) and the International Satellite Cloud Climatology Project HXG product on a three-hour scale (Tang et al., 2019) are generated by incorporating high-level satellite products and other ancillary data set. The unique Lagrange point orbit of Earth Polychromatic Imaging Camera (EPIC) onboard the Deep Space Climate Observatory (DSCOVR) are also utilized to

generate hourly DSR and PAR (Hao et al., 2018, 2019). Geostationary sensors provide new opportunities for estimating the DSR and PAR. Previous studies have successfully estimated surface shortwave radiation from geostationary sensors, such as multifunctional transport satellites (Huang et al. 2011; Li et al., 2015) and MSG Spinning Enhanced Visible and InfraRed Imager (Schmetz et al. 2002). With the launch of new-generation geostationary satellites, more studies have shifted to the Advanced Himawari Imager (AHI), Advanced Baseline Imager (ABI), and Advanced Geosynchronous Radiation Imager (AGRI), which provide higher spectral, spatial, and temporal resolutions with geometric and radiometric accuracies comparable to those of their polar-orbiting counterparts. Zhang et al. (2020) estimated the instantaneous and daily aggregated DSR from both AHI and ABI data using an optimization method with RMSE at $104\ W/m^2$ and $25\ W/m^2$ respectively. Letu et al. (2022) claimed a new benchmark of radiation datasets from AHI aerosol and cloud products with hourly DSR retrieval accuracy at RMSE of $104.9\ W/m^2$ and daily at $31.5\ W/m^2$. A recent study that employed a machine learning method to estimate half-hourly DSR achieved a validation RMSE of approximately $67\ W/m^2$, but their validation sites were limited and depended on the training sites (Chen et al., 2021). The resolution and accuracy of existed products still have space for improvements. Moreover, no uniform products based on multiple new-generation geostationary satellites are currently available, although several individual products have been produced by various agencies. Some products do not provide operational continuous data records or are not easily accessible to the international users. The map projection and data structure of full disk data files are also inconvenient for product processing and analysis.

The GeoNEX enhanced collection of new-generation geostationary satellite data across the globe makes it possible to produce a gridded high spatiotemporal resolution product of DSR and PAR with substantially improved quality. Through the GeoNEX platform, the data from various satellite sensors are preprocessed and archived in a consistent global tile gridding system. The GeoNEX data processing includes several critical data-improvement steps, such as removing the residual geometric errors, applying the orthorectification correction, and computing the precise view geometry at the individual pixel level. Based on the improved GeoNEX L1G TOA reflectance data, an operational high spatiotemporal resolution product of land surface DSR and PAR was produced. The physics-based look-up table (LUT) approach was selected as the retrieval algorithm, mainly because it is mature and reliable and has been used to generate the MODIS surface shortwave radiation product (MCD18). Besides, the algorithm does not require atmospheric products, such as cloud mask, aerosol and cloud optical properties as input, which are not currently available in the GeoNEX processing chain. The use of the high-quality input data and the mature algorithm created the foundation for estimating DSR and PAR with substantially improved quality and accuracy.

This data description paper introduces the estimation algorithm and procedures, summarizes the validation results of the new product and demonstrates its use in better understanding spatiotemporal variability of surface shortwave radiation. The remainder of this paper is organized as follows: Section 2 introduces the data and methods used to develop the GeoNEX DSR/PAR product; Section 3 presents validation and comparison results for the new data product; Section 4 uses two examples

to demonstrate the application of the high spatiotemporal resolution product in investigating DSR variability; Section 5 describes the data products and access information. Finally, Section 6 concludes the study with a summary.

## 2 Data and method

### 2.1 Method

This physics-based retrieval algorithm has been initially developed for the operational NASA MODIS DSR and PAR product (MCD18) (Liang et al., 2006; Wang et al., 2020). This algorithm has distinct advantages compared with other LUT methods. It mainly uses the blue band of TOA reflectance which is available for most sensors and less dependent on the additional atmospheric data which all ensure the high resolution, transferability, and continuity of the GeoNEX DSR algorithm. The extensive quality assessment of MCD18 showed that this algorithm is reliable, efficient, and highly accurate (Wang et al., 2021; Li et al., 2021).

To estimate surface shortwave radiation fluxes, two parameterization schemes that model the radiative transfer process between the atmosphere and the Lambertian surface were introduced (Liang, 2004; Chandrasekhar, 1960). The first parametrization scheme (Eq. 1) builds the relationship between the TOA spectral reflectance $R$, surface spectral reflectance $r$, and three parameters related to atmospheric conditions from the clearest condition to the cloudiest conditions at a given viewing geometry: path reflectance $R_0(\lambda)$, atmospheric spherical albedo $\rho(\lambda)$, and transmittance $\gamma(\lambda)$ for the spectral band:

$$R(\lambda) = R_0(\lambda) + \frac{r(\lambda)}{1-r(\lambda)\rho(\lambda)} cos(\theta_s)\gamma(\lambda)/\pi \quad (1).$$

The second parameterization scheme (Eq. 2) estimates the surface broadband radiation flux $F$ from surface reflectance $r$ and three atmospheric parameters: path irradiance $F_0$, atmospheric spherical albedo $\rho$, and atmospheric transmittance $\gamma$ at given $\theta_s$. $E_0$ is the extra-terrestrial solar broadband irradiance, which is adjusted by the distance to the sun:

$$F = F_0 + \frac{r\rho}{1-r\rho}E_0 cos(\theta_s)\gamma \quad (2).$$

The parameters used in the two equations are pre-calculated from the offline simulations with the atmospheric radiative transfer model. The results are saved in two LUT files. Two major steps are used to produce the GeoNEX DSR and PAR product from TOA reflectance data (Figure 1). In the first step, the first equation together with the TOA LUT file is used to predict the visibility index of the atmosphere. The geometric and radiometric corrected GeoNEX TOA reflectance values are the main input of this step. Total precipitable water vapor (TQV) is included to account for the effects of water vapor absorption. Surface elevation is also used to consider the effect of air mass on gas scattering and absorption. It should be noted that the terrain effect plays increasingly important role for estimating surface solar radiation at finer spatial resolution. The impacts of aspect

and slope on solar radiation need to be included in the future improvement. Given the viewing geometry (solar zenith angle (SZA $\theta_s$), view zenith angle (VZA), and relative azimuth angle (RAA)), possible values of TOA reflectance are calculated with Eq. 1 for various levels of visibility index (from the most clear atmosphere to the most cloudy case). The visibility index that provides the closest match between the calculated TOA reflectance and the observed TOA reflectance is retrieved. With the visibility index retrieved from the first step, surface shortwave radiation can be easily calculated from Eq. 2 and the surface LUT file through the second step.

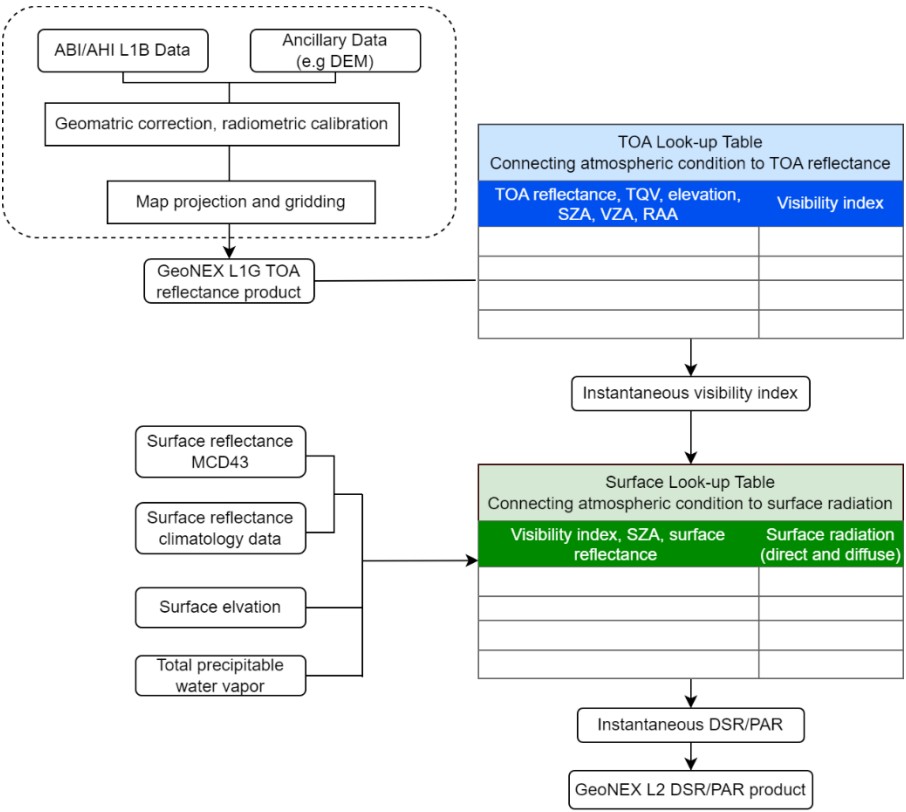

**Figure 1. The flowchart of generating the GeoNEX DSR and PAR product.**

## 2.2 Data

### 2.2.1 Input data

The input variables and corresponding data sources are listed in Table 1. The TOA reflectance data are obtained from the National Aeronautics and Space Administration (NASA) Advanced Supercomputing (NAS) Division GeoNEX platform, which archives a collection of gridded data from multiple geostationary satellites with consistent file formats and map

projections (Wang et al., 2020). The ABI onboard the GOES-16 geostationary meteorological satellites is equipped with 16 spectral bands. It produces full disk scanning every 10 or 15 min covering the area from 60°N, 138°W, to 60°S, 78°W. The AHI onboard Himawari-8 has 16 spectral bands. It produces full disk scanning every 10 min, covering regions from 60°N, 78°E to 60°S, 162°W. The blue band with a central wavelength of 0.47 um and 1 km spatial resolution for both ABI and AHI was used in this study to infer the atmospheric visibility index.

The MCD43 and surface reflectance climatology products provided surface albedo information. The surface reflectance was obtained from the MODIS product (Schaaf & Wang, 2015). Climatological surface reflectance data were used when no valid MODIS observation was available (Jia et al., 2022). The TQV data were obtained from the MERRA2 reanalysis product (Global Modeling and Assimilation Office, 2015) and the surface elevation data were from the Global 30 Arc-Second Elevation (GTOPO30) product (EROS, 2017).

**Table 1. The summary of input data**

| Name | Variable | Spatial Resolution | Temporal Resolution |
|------|----------|--------------------|--------------------|
| AHI/ABI TOA reflectance | TOA reflectance | 1km | 15min or 10min |
| View geometry | Solar zenith angle, sensor zenith angle, relative azimuth angle | 1km | 15min or 10 min |
| MCD43C3 | Surface albedo | 0.05 degree | Daily |
| MERRA2 | Total precipitable water vapor | 0.5 x 0.625 degree | Hourly |
| GTOPO30 | Surface elevation | 30 arc seconds | Static |
| Surface reflectance climatology | Surface reflectance | 0.05 degree | Static/daily |

**2.2.2 Validation data**

Measurements from 63 stations in the ABI or AHI spatial domain were collected to validate GeoNEX DSR/PAR products. Among these, 34 sites were located in the ABI coverage domain and 29 sites in the AHI coverage area (Figure 2). The stations belong to four networks, with 25 sites from AMERIFLUX, 11 sites from the Baseline Surface Radiation Network (BSRN), 20 sites from FLUXNET, and 7 sites from the Surface Radiation Network (SURFRAD). All 63 sites had DSR data. The processes of ground measurement data quality checks including daily and monthly aggregation follow Li et al. (2021). Only 27 sites recorded PAR measurements. Seven sites from SURFRAD measure the PAR flux directly while the rest sites from AMERIFLUX record PAR data in quantum units (photosynthetic photon flux density, $\mu mol\ m^{-2}s^{-2}$). The conversion between the quantum units to the energy units follows Dye (2004).

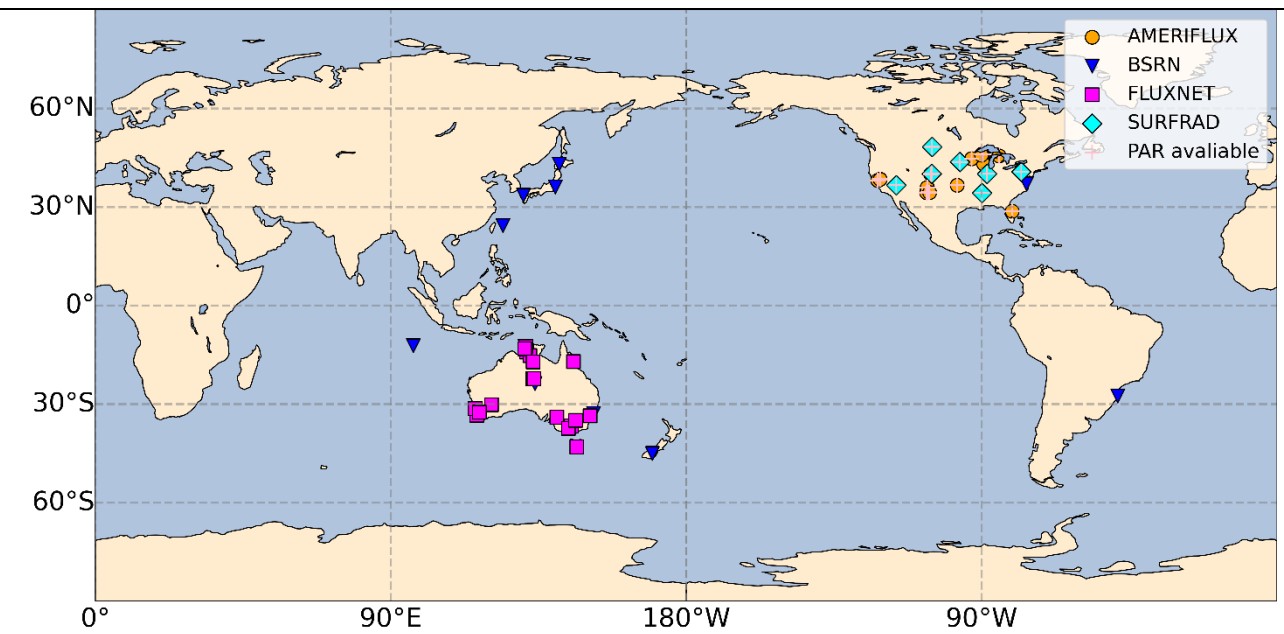

**Figure 2. The map of ground stations and their networks used for validating the GeoNEX DSR/PAR product.**

### 2.2.3 Other products

Existing DSR products were used in this study for comparison with the new GeoNEX product. The EPIC/DSCOVER generates DSR and PAR globally at 0.1° spatial resolution and 1-2h temporal resolution. A random forest approach was applied to the data obtained by the EPIC sensor onboard the DSCOVER (Hao et al., 2018, 2019). The CERES SYN1deg product provides hourly, 3hourly, daily, and monthly radiation data at the surface, TOA, and various atmospheric layers with a 1° spatial resolution. The data on surface shortwave fluxes were calculated with Langley Fu-Liou radiative transfer code from polar-orbit and geostationary satellite data, as well as other ancillary information (Rutan et al., 2015). CERES-SYN data has been extensively validated in previous studies, showed the highest accuracy compared with the most existing products; hence, it has been widely used as a baseline product (Riihelä et al., 2017; Sun et al., 2018; Li et al., 2021). The National Oceanic and Atmospheric Administration (NOAA) GOES-R series Level 2 product (ABI-L2-DSR) was also evaluated in this study. The full disk data were produced on a global latitude/longitude grid at 0.5° resolution, employing two retrieval path methods to estimate DSR (GOES-R Algorithm Working Group and GOES-R Program Office, 2017). A new version of MCD18A1 has also been included in the comparison (Wang et al., 2020). It applied a similar LUT method over MODIS TOA reflectance and estimated DSR at a global scale at 1 km spatial resolution with 3hourly and daily interpolated temporal resolutions.

In addition, this study employed NOAA GOES-R Series Level 2 clear sky mask data to investigate the performance of various DSR/PAR products under different cloud conditions. The product contains images in the form of binary cloud masks. To match

it with hourly DSR products, the 15 min full disk data were aggregated. The sample was classified as under cloudy conditions if all four observations in an hour were cloudy and as clear if all four observations within an hour were clear. The rest were classified as partial cloudy conditions.

## 3 Quality assessment and error analysis

### 3.1 Overall validation

The GeoNEX DSR and PAR product is currently produced from the GOES16-17/ABI and Himawari8/AHI data, covering the area of East Asia, Australia, North America and South America from 78°E to 18°W between 60°N and 60°S (Figure 3). The GeoNEX hourly and daily DSR data were validated with ground measurements over one year to provide a comprehensive evaluation of the product across various seasons (Figure 4). The $R^2$ values for the hourly DSR from ABI and AHI were 0.929 and 0.950, respectively. RMSE were 78.2 and 69.4 $W/m^2$ and relative RMSE (rRMSE) were 19.7% and 16.2%. After aggregating to the daily values, the uncertainties in estimating DSR were further reduced, while the $R^2$ increased to 0.968 and 0.972 for ABI and AHI, respectively. The RMSE (rRMSE) achieved 18.4 (9.6%) and 17.2 (8.3%) $W/m^2$. To the best of our knowledge, it is the first satellite product of DSR with the rRMSE lower than 10% (Li et al., 2021). The validation accuracies over ABI and AHI coverage are similar, with slightly better accuracy over AHI. This is partly due to the more homogeneous and constant atmospheric conditions in the AHI domain. After aggregating into daily intervals, the differences between the two sensors decreased. The accuracy of the DSR estimation varies with cloud conditions. As shown in Figure 5, the rRMSE over cloudy sky (30.6%) is triple that over clear sky (10.9%), and the rRMSE of partial cloud samples (18.3%) fell in the middle of accuracies under clear and cloudy skies. The elevated errors for cloudy-sky cases partly originate from the assumption of homogenous and plane-parallel clouds in the radiative transfer code (Chen et al., 2019; Van Laake & Sanchez-Azofeifa, 2004). The linear interpolation processes in searching through the two LUTs may also lead to uncertainties in the results.

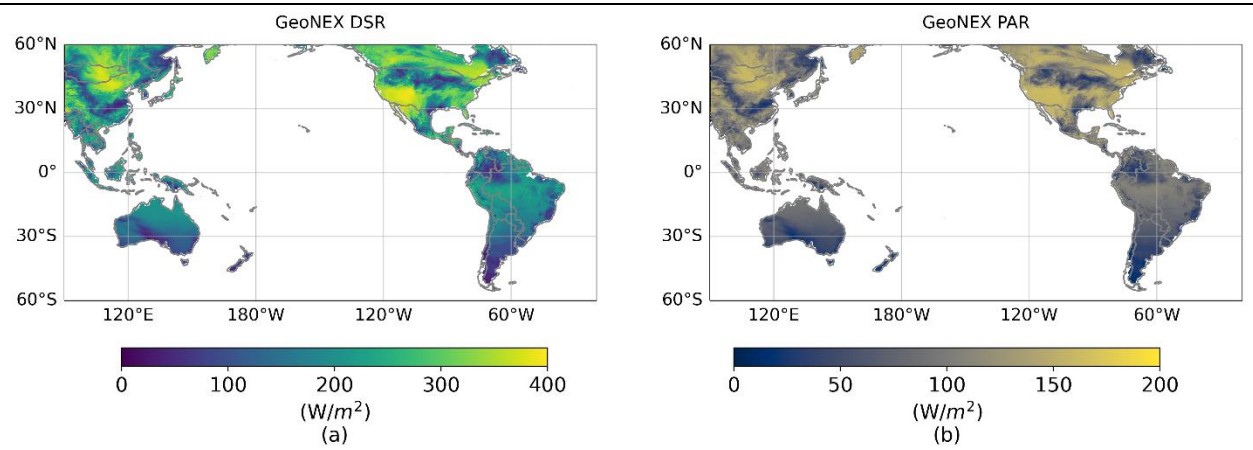

**Figure 3. The daily DSR (a) and PAR (b) maps from the GeoNEX product on June 19th 2018.**

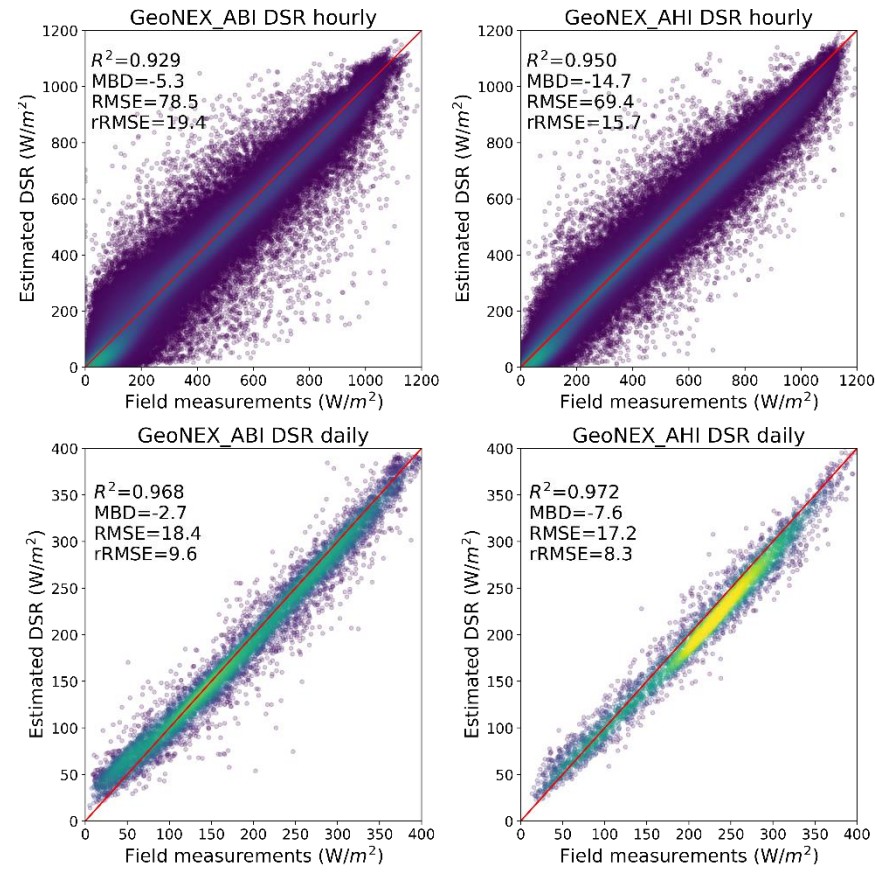

195

**Figure 4. Scatter plots of the estimated and observed daily and hourly DSR values for ABI and AHI data.**

Figure 6 presents the validation results of the GeoNEX PAR estimation. The $R^2$ for hourly PAR estimation was 0.927, and the RMSE (rRMSE) was 34.7 (19.7%) $W/m^2$. The $R^2$ for daily estimation was 0.956, and the RMSE (rRMSE) was 9.5 (10.8%) $W/m^2$. Since the PAR values of some sites are converted from the photosynthetic photon flux density which has systematic uncertainties (Dye, 2004), we also include the validation results only over SURFRAD sites where PAR flux is provided directly. The accuracies of both hourly and daily PAR estimations were higher than those of the existing products and experimental studies (Li et al., 2015; Hao et al., 2018).

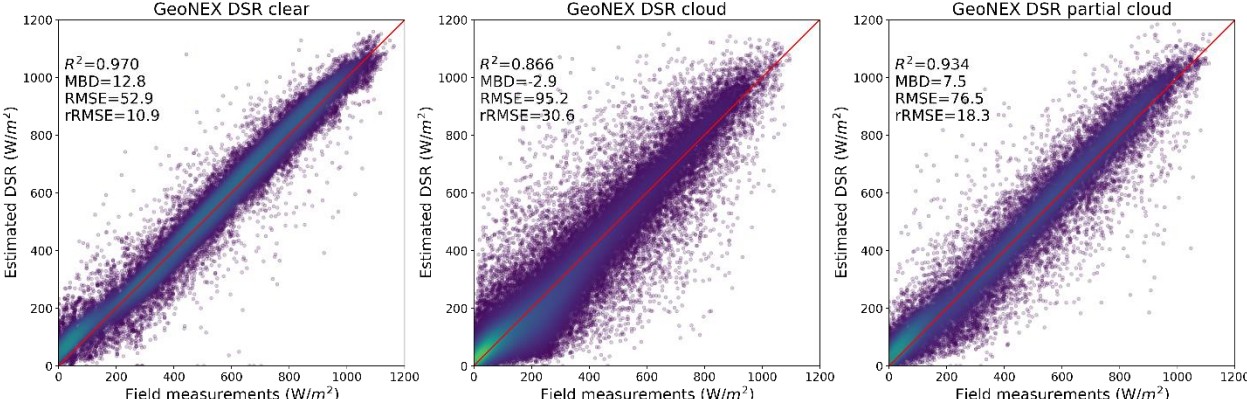

**Figure 5. Scatter plots of the estimated and observed hourly DSR values for ABI coverage under different cloud conditions.**

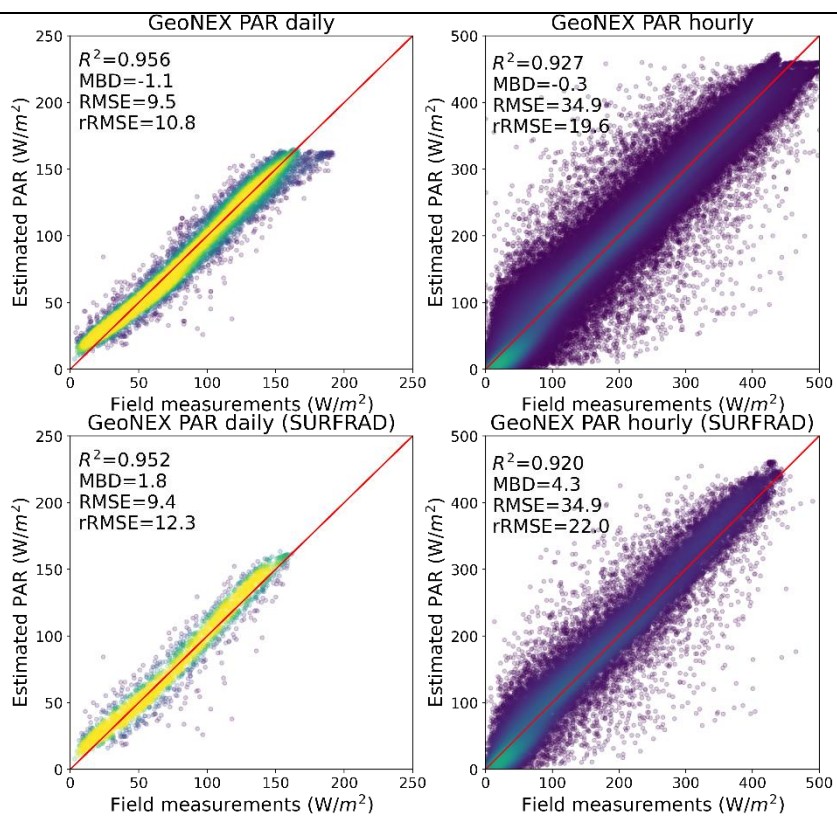

**Figure 6. Scatter plots of the estimated and observed daily and hourly PAR values for ABI coverage.**

## 3.2 Comparison with existing products

We also compared the new GeoNEX DSR product with four existing DSR products at hourly and daily scales over identical samples. The statistics are summarized in Table 2. The correlation between the CERES hourly DSR product and the ground measurements had an $R^2$ value of 0.904 and RMSE of 91.6 $W/m^2$, followed by EPIC/DSCOVER with $R^2 = 0.798$ and RMSE = 130.8 $W/m^2$. ABI-L2-DSR had an $R^2$ value of 0.748 and RMSE of 148.8 $W/m^2$. The proposed GeoNEX DSR outperformed all existing products (Letu et al., 2022; Zhang et al., 2021; Hao et al., 2019) with $R^2$ of 0.928 and RMSE of 78.3 $W/m^2$. The relatively lower performance of the ABI-L2-DSR data may be partly because the ABI-L2-DSR is an instantaneous estimation and has a coarse spatial resolution of 0.5 °. Table 3 presents a comparison of hourly products under different cloud conditions. CERES and GeoNEX achieved comparable accuracy under clear-sky conditions. Over cloudy skies, the GeoNEX product exhibits superior accuracy with an RMSE of 95.2 $W/m^2$. The RMSE of CERES and EPIC were as high as 112.8 and 159.0 $W/m^2$, respectively.

For daily estimation, MCD18, which retrieves DSR from the polar orbiting sensor at the highest spatial resolution of 1 km,

was also included in comparison with CERES and EPIC. Other mature daily DSR products, such as GLASS, CLARA, and BESS, were not included in this study because they have shown comparable or inferior performance to CERES (Li et al., 2021). Similar to the hourly results, GeoNEX DSR outperformed all existing datasets with $R^2$ of 0.965 and RMSE of 18.9 $W/m^2$.

We emphasize that the GeoNEX DSR/PAR algorithm was not trained or tuned with the field measurements used for comparison. The validation and comparison results show that this new GeoNEX shortwave radiation product provides a highly accurate DSR estimation from satellites, with hourly RMSE lower than 80 $W/m^2$ and daily RMSE lower than 20 $W/m^2$.

**Table 2. Summary of the comparison results between the GeoNEX DSR product and other DSR products**

| Product | $R^2$ | BIAS ($W/m^2$) | RMSE ($W/m^2$) |
|---------|-------|----------------|----------------|
| | | Instantaneous | |
| ABI-L2-DSR | 0.75 | -13.9 | 148.8 |
| | | Hourly | |
| CERES | 0.90 | 2.7 | 91.6 |
| EPIC | 0.80 | 2.8 | 130.8 |
| GeoNEX | 0.93 | -3.4 | 78.3 |
| | | Daily | |
| CERES | 0.94 | 3 | 24.5 |
| MCD18 | 0.91 | -5.2 | 32.6 |
| ABI-L2-DSR | 0.77 | -7.4 | 48.1 |
| EPIC | 0.83 | 5.9 | 41.5 |
| GeoNEX | 0.97 | -2.4 | 18.9 |

**Table 3. Comparison of the GeoNEX DSR product with other DSR products under different cloud conditions**

| Product | $R^2$ | BIAS ($W/m^2$) | RMSE ($W/m^2$) | $R^2$ | BIAS ($W/m^2$) | RMSE ($W/m^2$) | $R^2$ | BIAS ($W/m^2$) | RMSE ($W/m^2$) |
|---------|-------|----------------|----------------|-------|----------------|----------------|-------|----------------|----------------|
| | | Clear | | | Cloud | | | Partly cloud | |
| CERES | 0.97 | -11.1 | 52.4 | 0.82 | 19.0 | 112.8 | 0.91 | -10.1 | 91.8 |
| EPIC | 0.94 | -45.8 | 86.2 | 0.69 | 50.7 | 159.0 | 0.85 | -15.6 | 115.9 |
| GeoNEX | 0.97 | -12.8 | 52.9 | 0.87 | 2.9 | 95.2 | 0.93 | -7.5 | 76.5 |

To evaluate their capability to monitor the temporal variability of DSR, the diurnal cycles of GeoNEX, EPIC, and CERES DSR estimations were plotted together with in-situ measurements at seven SURFRAD sites in June 2018 (Figure 7). Although all three products could depict the diurnal trends of DSR, their performances diverged substantially over the days with high 235 DSR variability (i.e., days of year (DOY) 174 and 175) and mountainous areas (i.e., TBL). EPIC was prone to overestimation when potential clouds existed. CERES agreed well with the in-situ measurements, but could not capture the sharp changes in DSR as accurately as GeoNEX due to coarse spatial resolution of CERES data.

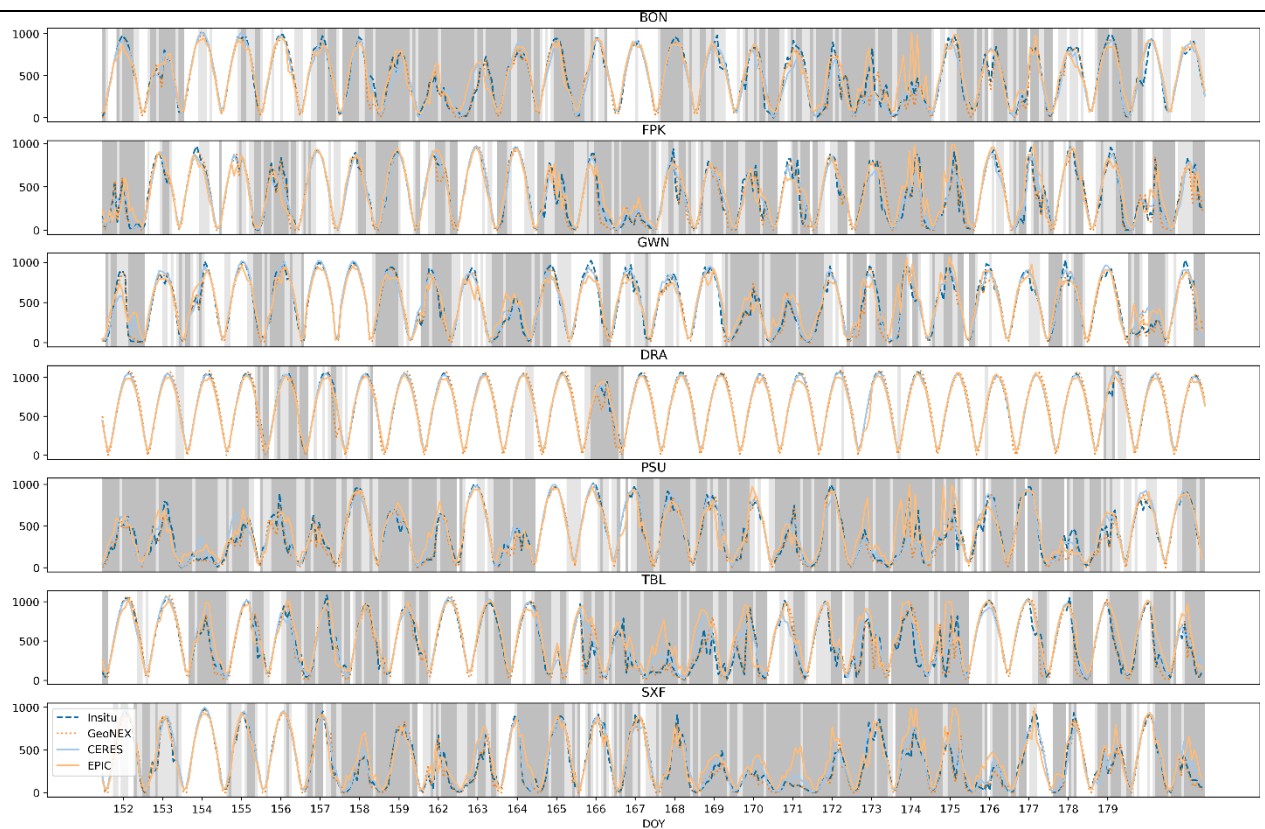

**Figure 7. The diurnal cycles of GeoNEX, EPIC, and CERES estimations compared with in-situ measurements at seven SURFRAD sites in June 2018. Cloudy conditions are marked in dark grey and partially cloudy conditions are in light grey.**

### 3.3 Impact of viewing geometry on estimation errors

The diurnal variation of the GeoNEX DSR estimation was examined and is presented in Figure 8. The hourly averaged DSR matched well with the in-situ measurements for both ABI and AHI. No noticeable changes in the RMSE and BIAS were observed throughout the day. However, owing to the small average DSR values at the start and end of a day, the rRMSE increases dramatically. This phenomenon occurs in most DSR products, partly due to the Lambertian assumption adopted in the radiative transfer model.

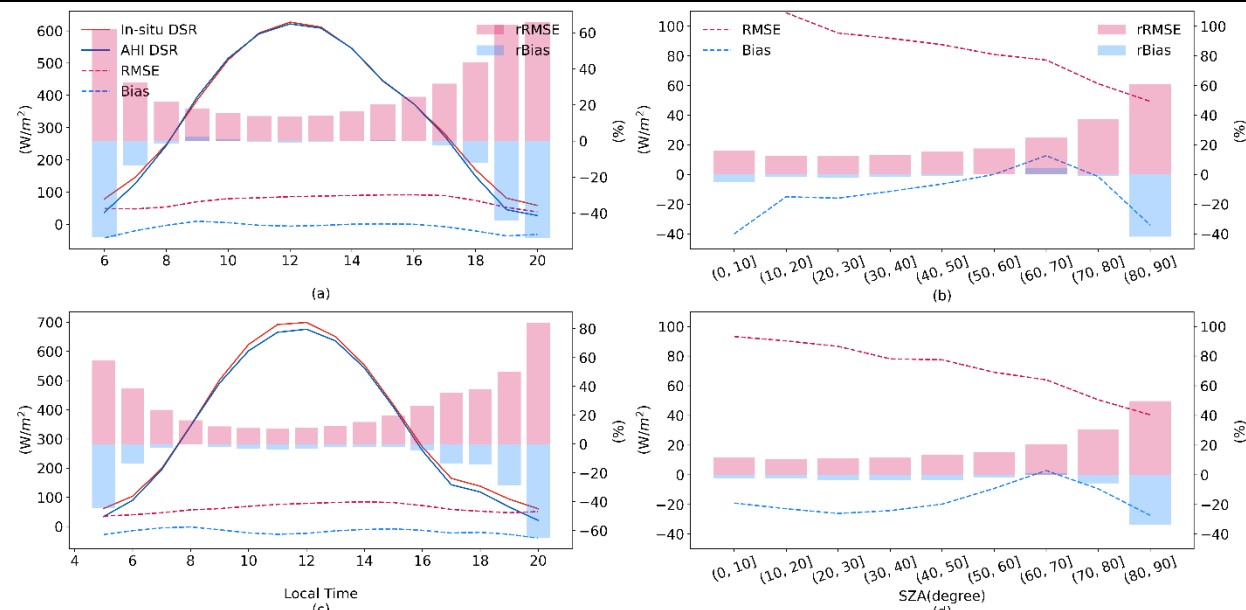

**Figure 8: Diurnal variation of DSR estimation for ABI (a) and AHI (c), and the impact of SZA on ABI (b) and AHI (d) estimation. The left-hand y-axis corresponds to the line plot, while the right-hand y-axis corresponds to the bar plot.**

Geostationary satellites maintain a static position relative to the Earth, and thus each pixel in the image has a fixed value of VZA. Figure 9 presents the rRMSE and rBias of each site located under AHI and ABI coverage. The pink stars represent the positions of the AHI and ABI sensors. For the rRMSE, a radial distribution is presented. A large rRMSE existed at sites far from the sensors. The same result was obtained for the rBias distribution. Underestimation was observed over the sites near the sensor, whereas overestimation was observed for sites far from the sensor. Overall, more uncertainties may exist at higher latitudes, as the geostationary sensors are located at the equator. To quantitatively analyze the influence of VZA, regression lines between VZA and rRMSE/rBias are plotted in Figure 10. Positive slopes exist for both rRMSE and rBias. The p-values for both the rRMSE and rBias are less than 5%, which demonstrates that these positive correlations are significant.

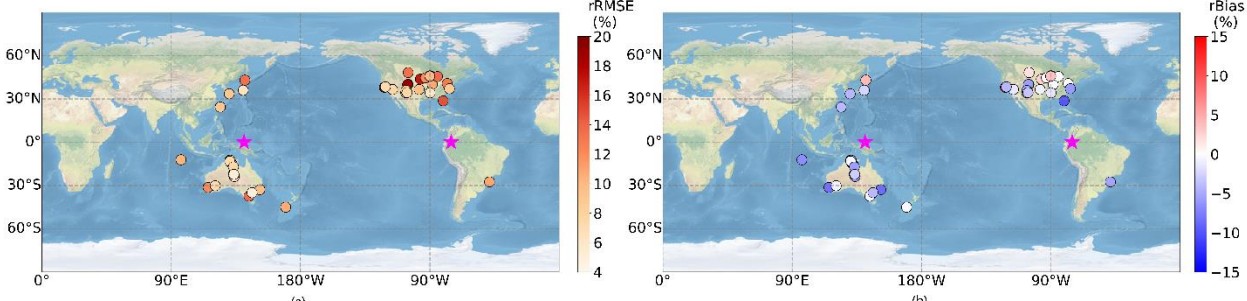

**Figure 9. The spatial distribution of rRMSE (relative RMSE) and rBias (relative bias) of DSR estimation. The pink stars**

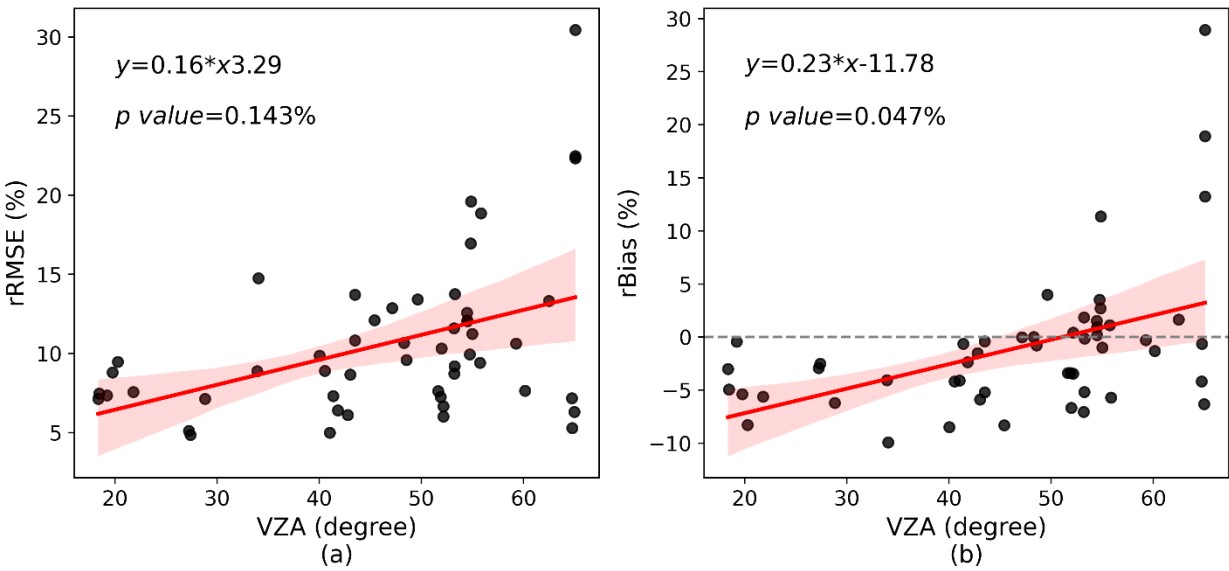

**represent the position of the AHI and ABI sensors.**

**Figure 10. The impact of VZA on DSR estimation rRMSE (a) and rBias (b). The regression equation and p-value are shown.**

**3.4 Impact of spatial and temporal resolutions on estimation errors**

Previous studies have suggested that the accuracy of DSR estimation is influenced by the spatial and temporal aggregation scales (Li et al., 2021; Zhang et al., 2021). For instantaneous PAR and DSR estimation, the optimal scale for applying 1-D transfer models is approximately 20 km (Chen et al., 2019; Zhang et al., 2021). For daily estimation, Li et al. (2021) demonstrated that generally lower spatial resolution can result in higher accuracy for most existing products, but it is noticeable that the products validated in previous studies are usually interpolated from instantaneous estimation. To further examine the influence of spatial and temporal resolution on surface shortwave radiation estimation, this study compared the accuracy of DSR estimation at different scales (Figure 11 and Table 4). This agrees well with previous findings that temporal aggregation exerts a greater impact on accuracy than spatial aggregation (Zhang et al., 2021). The hourly rRMSE was approximately 18%, while the monthly are approximately 6%. The higher the temporal resolution, the greater the influence of the spatial resolution on the estimation accuracy. As shown in Figure 11, the differences in RMSE among different spatial resolutions decreased as the temporal resolution decreased. At the hourly scale, the highest rRMSE reached 19.8% at 100 km and the lowest was 17.1% at 10 km, whereas at the monthly scale, the DSR estimation was nearly independent of the spatial scale. Moreover, compared with previous analysis of instantaneous interpolated daily DSR estimation (Li et al., 2021), our results at a daily scale are less variable among different spatial scales, suggesting that the aggregation of hourly DSR is a possible solution to mitigate the impact of spatial resolution on daily DSR estimation and enables daily DSR estimation at spatial resolutions as high as 1 km.

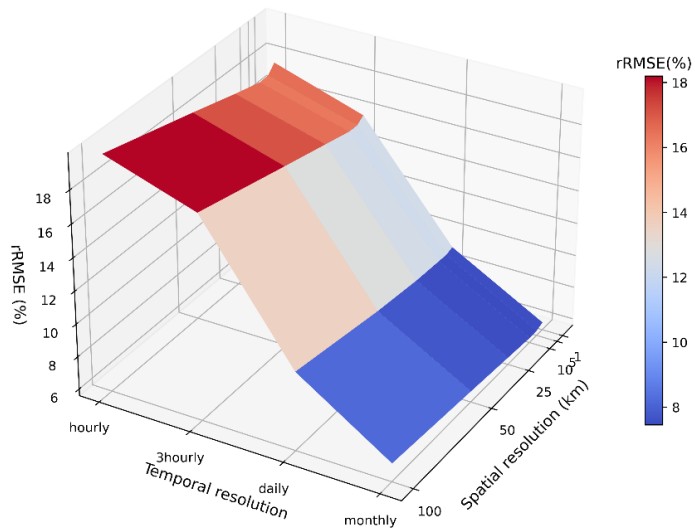

**Figure 11. Influence of spatial and temporal resolution on DSR estimation.**

**Table 4. Summary of spatial and temporal influence on hourly and daily DSR estimation**

| Resolution | $R^2$ | BIAS $(W/m^2)$ | RMSE $(W/m^2)$ | $R^2$ | BIAS $(W/m^2)$ | RMSE $(W/m^2)$ |
|---|---|---|---|---|---|---|
| | | Hourly | | | Daily | |
| 1km | 0.94 | -9.4 | 74.3 | 0.97 | -4.3 | 18.0 |
| 5km | 0.94 | -9.8 | 72.2 | 0.97 | -4.4 | 17.7 |
| 10km | 0.94 | -9.3 | 72.0 | 0.97 | -4.4 | 17.8 |
| 25km | 0.94 | -8.8 | 73.9 | 0.97 | -4.1 | 18.0 |
| 50km | 0.93 | -8.7 | 77.2 | 0.97 | -4.1 | 18.6 |
| 100km | 0.92 | -8.4 | 83.8 | 0.96 | -4.0 | 20.4 |

### 3.5 Analysis of sensitivity to the input parameters

Surface reflectance and TQV are two major ancillary input variables to the DSR retrieval algorithm. To study the impact of input data quality on retrieval errors, a sensitivity study was conducted with the simulated data. Random errors of various levels were added to the two input variables. The error-added surface reflectance and TQV were then used to estimate DSR

through the LUT retrieval approach. The estimated DSR were compared with the true DSR previously simulated to calculate retrieval errors. The mean absolute error (MAE) were then calculated for various levels of input errors (Figure 12). For both input variables, MAE of estimating DSR increases with the errors in input data. In general, the DSR accuracy is dependent more on surface reflectance than TQV, because the LUT algorithm needs the difference information between surface

reflectance and TOA reflectance to infer atmospheric conditions. For example, 10% of error in surface reflectance can lead to a MAE increase around 10 $/m^2$, while for TQV it is about 0.5 $W/m^2$. The results also suggest the sensitivity to the two input variables both change with solar angles. Larger MAE is generated when SZA is higher. However, the retrieval errors increase much more quickly with SZA for the input variable of TQV than surface reflectance. When SZA is within 60 to 80 degrees, the MAE can achieve near 2.5 times of the MAE when SZA is lower than 20 degrees. It can be attributed to the stronger absorption of water vapor as SZA increases.

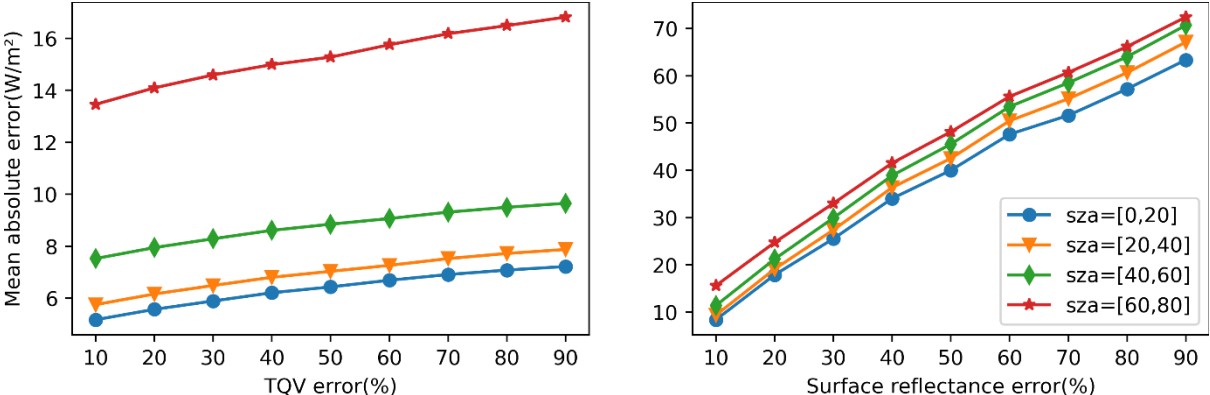

**Figure 12. Sensitivity of the DSR retrieval algorithm to errors in the input data of surface reflectance and TQV.**

## 4 Application demonstrations

The importance of spatial and temporal heterogeneity of the DSR has been demonstrated in many studies (Gueymard et al., 2011; Yan et al., 2018; Sweerts et al., 2019). However, such issues have not been fully investigated owing to the limited spatial and temporal resolutions as well as relatively low accuracy of the existing products. The new GeoNEX DSR/PAR product, with their unique characteristics, provide a valuable opportunity to re-examine these issues. Here, two examples are used to demonstrate the applications of the high spatial-temporal resolution DSR product. In Section 4.1, we investigated how the overpass time and counts of polar-orbiting satellites affect the accuracy of estimating the daily DSR values. In Section 4.2 we studied the spatial heterogeneity of DSR at various temporal scales.

### 4.1 Effects of overpass time on estimating daily DSR

As shown in Table 2, although the GeoNEX DSR and PAR adapted the heritage algorithm of the MCD18, its accuracy is much higher than that of MCD18 because of the high temporal resolution of geostationary data. One reason is that the quality of existing DSR products derived from polar-orbiting satellite data relies heavily on temporal upscaling schemes to calculate the

daily DSR from instantaneous observations (Wang et al., 2010). We took advantage of the high frequency of the GeoNEX

DSR product to simulate how the estimates of daily DSR change with overpass time and counts.

The modislike11 and modislike13 data are generated using visibility indexes at local times 11:00 and 13:00, corresponding to the Terra and Aqua passing time, as the constant atmospheric condition of the whole day. The Modislike2p data were generated to emulate the cases where observations from both Terra and Aqua are available. It uses the visibility index at 11:00 to represent the atmospheric condition before 11:00 and the visibility index at 13:00 to represent that after 13:00. Between 11:00 and 13:00, The visibility index was linearly interpolated (Wang et al., 2010). A similar interpolation method was used to generate the MCD18 products (Wang et al., 2020). The mechanism is shown in Figure 13, where different data were compared over the seven SURFRAD stations on June 19th. The results show that all methods work well under no-cloud conditions, such as at the DRA site, where the visibility index at 13:00 and 11:00 can represent the entire day's atmospheric condition. However, large uncertainties arise when atmospheric conditions vary substantially during the day. Such uncertainties may be reduced with additional observations in the modislike2p scenario, as in the case of the PSU. Nevertheless, the limited number of observations is one of the major error sources for estimating the daily DSR from polar-orbiting satellite data.

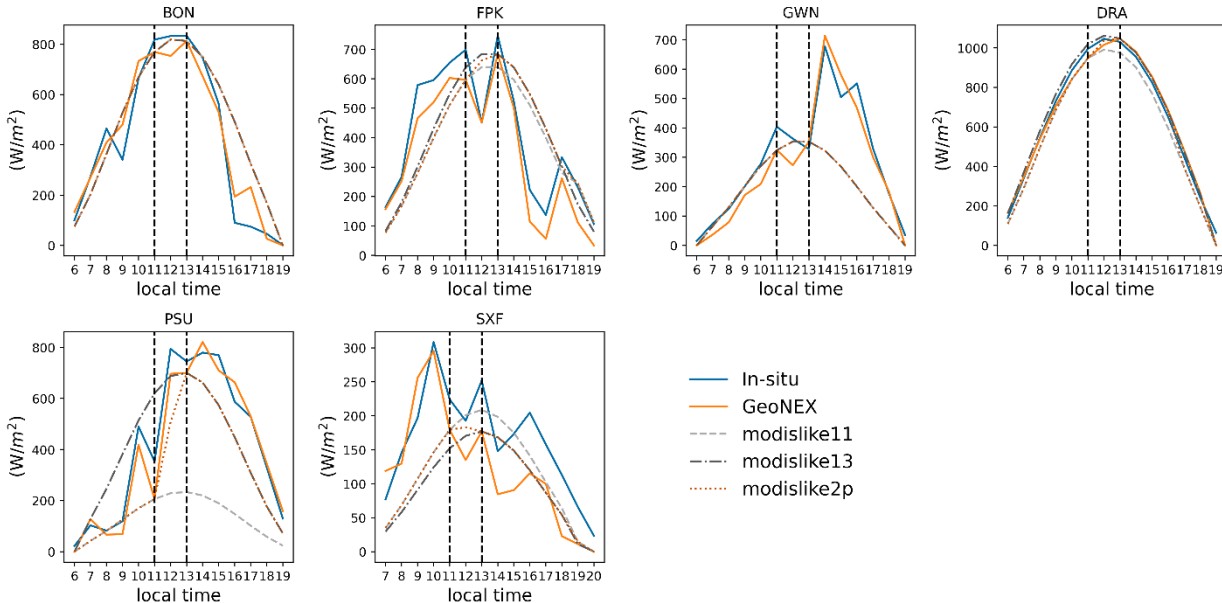

**Figure 13. Comparison of hourly DSR between the GeoNEX product and the MODIS-interpolated data over seven SURFRAD stations on June 19th.**

The relative bias (rBias) and RMSE (rRMSE) of the daily averaged DSR between modislike interpolated data and ABI data throughout the year were calculated (Figure 14). The rBias maps showed that the representativeness of the visibility index at

11:00 varied spatially. More overestimations were observed over high-elevation areas, and underestimations were observed at the edge of these areas, which may be because the morning clouds have not been formed or could not reach certain heights. Moreover, the visibility index at 13:00 h was not sufficiently representative. The rBias map shows that interpolation from the visibility index at 13:00 will lead to underestimation over all study areas, which is attributed to more cloud formation in the afternoon. The rRMSE maps demonstrate the efficiencies of incorporating two passes when interpolating daily DSR from polar-orbiting sensors, as the modislike2p generates less variability compared with ABI-based daily DSR; however, the average rRMSE reaches 10%. For both modislike10 and modislike 13, the high rRMSE is around the mountainous areas, the maximum rRMSE is approximately 70%, and the average rRMSE is approximately 18%.

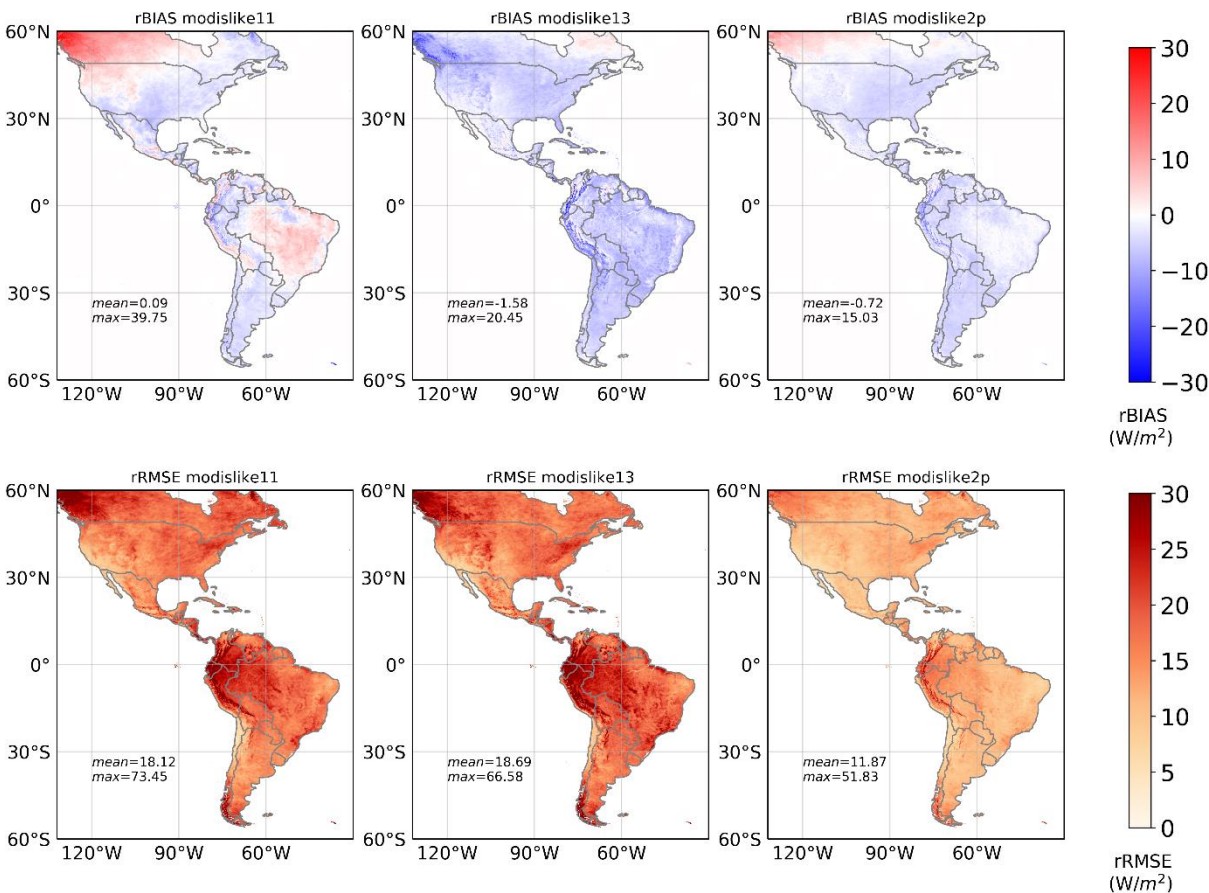

**Figure 14. The temporal representation maps showing the rBias (relative bias) and the rRMSE (relative RMSE) of daily-averaged DSR between the GeoNEX product and the MODIS-interpolated data from visibility index at local times of 11:00 (modislike11), 13:00 (modislike13), and both (modislike2p).**

## 4.2 Spatial heterogeneity of DSR

Existing global or regional shortwave radiation products mostly have a spatial resolution coarser than 5 km, which meets the requirements of some terrestrial models. However, other studies may require data with a much higher spatial resolution. For instance, whether the character of a solar resource at one location can be representative of nearby locations is a critical question for solar grid design and deployment. Attempts have been made to analyze the spatial heterogeneity of DSR, but most have focused on regional scales or used coarse resolution data as inputs (Kariuki & Sato, 2018; Sarr et al., 2021; Tapia et al., 2022). With the help of the high spatiotemporal resolution GeoNEX DSR data, we were able to quantify the spatial heterogeneity of DSR at a large spatial scale over different temporal scales.

We employed the metric used in the previous studies (Gueymard et al., 2011; Yan et al., 2018) to calculate the coefficient of variance (COV) to represent spatial heterogeneity of DSR. COV is defined as:

$$COV = \frac{\sigma_n}{E_n} * 100$$

where n denotes the number of pixels surrounding the central pixel. N was set as 10×10 and 100×100 km, respectively, to examine spatial heterogeneity at different scales. $\sigma_n$ and $E_n$ are the standard deviation and mean of these n pixels, respectively.

Annual and seasonal spatial representation maps were generated at 10×10 and 100×100 km, which highlight areas susceptible to high heterogeneity in the DSR (Fig. 14). A high COV usually corresponds to mountainous and high elevation areas. For the annual COV in the 10×10 matrix over CONUS, the lowest COV occurs in central Missouri and increases towards the east and west coasts. Some regions at the edge of the American Cordillera, such as Denver, have a high COV. A high COV extends from northern Rocky in Canada along the American Cordillera to Mexico and further reaches the entire Andes Mountains in South America. Over Asia and Oceania under the AHI coverage (Fig. 14), a high COV is present at the edge of the Tibetan Plateau, especially along the Himalayan Mountains and extends to the Annamite range. It also occurs in mountainous regions of island countries such as Indonesia, Japan, and New Zealand. The patterns between 10×10 km and 100×100 km were similar, with greater variability and extent in the latter representation maps. Larger COV pixels appear along the Appalachian Mountains in the US, eastern mountainous areas in Brazil, the northern part of the Tibetan Plateau in China, and southeast Australia. The analysis here suggests the need for high-spatial-resolution DSR data for these regions. Some seasonal changes in the COV values were also observed. We plotted aggregated June, July, and August (JJA) as well as December, January, and February (DJF) aggregated variation maps, as shown in Figure 15. In general, a higher variance was observed in the Northern Hemisphere during DJF and in the Southern Hemisphere during JJA. It impacts high latitude most. It is also noticeable that a horizontal line appears at approximately 55°N in both AHI and ABI 100×100km maps during DJF. This might correspond to the polar front where a sharp gradient in temperature occurs and suggests that these two air masses with different temperatures leads to significant DSR variation at the surface at a 100 km scale.

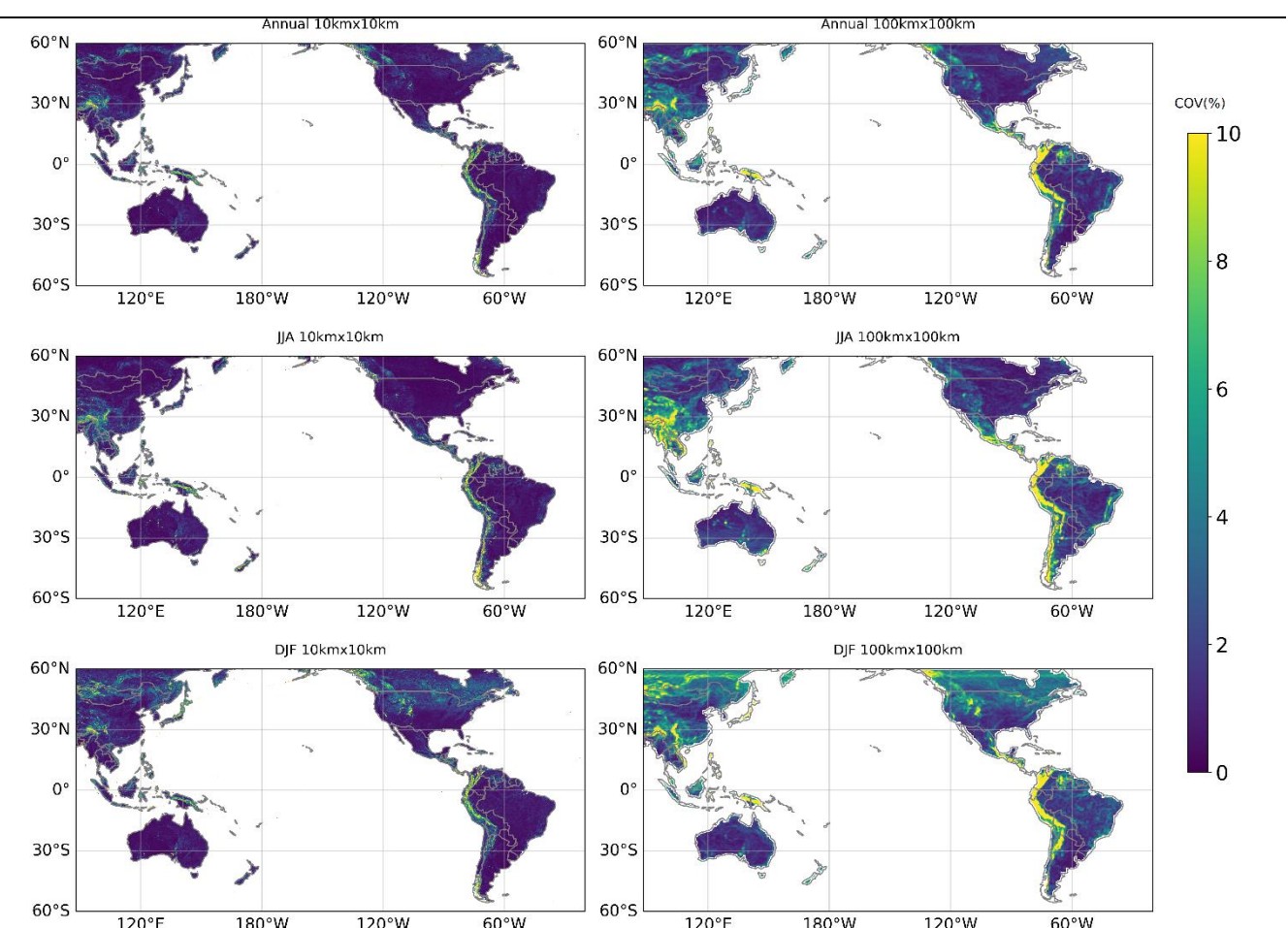

**Figure 15. Spatial coefficient of variance (COV) maps at 10kmx10km and 100kmx100km scales for annual; JJA (June, July, and August); and DJF (December, January, and February).**

Figure 16 shows the distribution of daily and hourly COV frequencies at 10 and 100 km in all study areas. For the daily COV at 100 km, 86% of the areas were within the range of 3%-10%. For daily COV at 10 km, 85% of areas showed a COV lower than 3%, and most areas had a COV distributed within the range of 2%-3%. For the hourly COV at 100 km, 73% of the areas had a COV higher than 5%, and most areas were distributed within 20%-30%. For the hourly COV at 10 km, 64% of the areas had a COV within the range of 3%-10%. The values between 5%-10% were dominant for COV. The results demonstrate that

the higher the temporal resolution, the more severe the spatial heterogeneity issues. DSR products with spatial resolutions of approximately 100 km are not sufficient for analysis at temporal scales higher than daily for most areas. The 10 km DSR products may be sufficient for analysis using daily DSR data, but for those with hourly data, uncertainties in COV of 5%-10% existed for most areas. The results further demonstrate the importance of high-spatial-resolution products, particularly at high

temporal resolution.

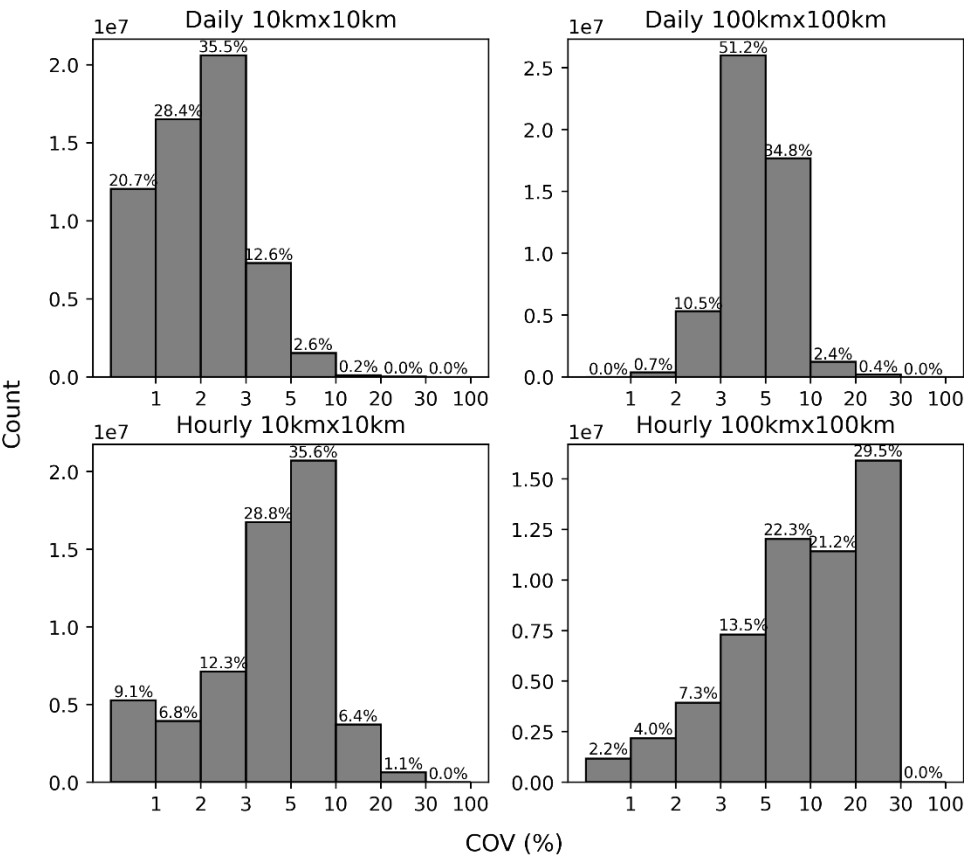

**Figure 16. Histogram of the coefficient of variance (COV) for daily and hourly scales at 10 km x 10 km and 100 km x 100 km resolutions over the study area. The number above each bar indicates the relative frequency.**

## 5 Data availability

The GeoNEX DSR and PAR product is a gridded GeoNEX Level 2 data set and can be accessed through the NASA GeoNEX data portal at https://data.nas.nasa.gov/geonex/geonexdata/GOES16/GEONEX-L2/DSR-PAR/ and https://data.nas.nasa.gov/geonex/geonexdata/HIMAWARI8/GEONEX-L2/DSR-PAR/

(https://doi.org/10.5281/zenodo.7023863, Wang & Li, 2022). The DSR and PAR data files are organized in the standard GeoNEX tile system with the geographic latitude/longitude projection. Each tile file has a dimension of 600 by 600 pixels at

a spatial resolution of 0.01°, covering a region of 6° by 6°. Detailed information regarding the GeoNEX tile system can be found at https://www.nasa.gov/geonex/dataproducts. The DSR/PAR data files include three scientific datasets: hourly DSR array, hourly PAR array, and quality control (QC) array. The DSR or PAR values should be multiplied by a scaling factor (0.1) before use. The QC was used to indicate the quality of DSR and PAR estimation (0: high quality retrieval using the observed surface reflectance data, 1: degraded data using the climatology reflectance data, 2: invalid retrieval and filling value).

## 6 Conclusions

Based on the GeoNEX enhanced L1G TOA reflectance data, an operational high spatiotemporal resolution product of DSR and PAR was produced for multiple new-generation geostationary satellite sensors. The GeoNEX DSR and PAR product can be freely accessed through the NASA GeoNEX data portal in a user-friendly global tile system. The new product also displays superior data quality due to the reliable retrieval algorithm and the high-quality input data. Compared to the original geostationary data, the data enhancement provided by the GeoNEX L1G processing includes removal of the residual geometric errors, the orthorectification correction and the supplement of precise view geometry information at the pixel level. The current GeoNEX collection includes GOES16, GOES17 and Himawari 8 satellite covering from 78°E to 18°W between 60°N and 60°S. The ultimate goal is to incorporate additional new-generation geostationary data to provide a global coverage between 60°N and 60°S.

The new GeoNEX DSR and PAR product was extensively validated against one year of ground measurements across multiple continents. The comparison with existing products demonstrated the superior accuracy of the GeoNEX DSR and PAR product. The RMSE of hourly DSR estimation is $74.3\ W/m^2$ (18.0%) and that of daily DSR estimation is $18.0\ W/m^2$ (9.2%) when evaluated against 63 sites from four different networks. The hourly PAR achieves $34.9\ W/m^2$ (19.6%) and daily PAR achieves $9.5\ W/m^2$ (10.5%) validated over 27 sites. It should be noted that the GeoNEX DSR and PAR data were retrieved using a physical LUT approach and did not require any training or tuning based on any of the validation sites.

The high-quality gridded dataset of surface incident shortwave radiation provides new opportunities to study its spatial and temporal variability. We demonstrate the application of this new product using two examples. We first mapped the errors in estimating the daily DSR from the polar-orbit satellite data. It was found that one observation per day led to an average relative RMSE of 18 %, and an increase in the daily observation number to two reduced the relative RMSE to 10%. In addition, we characterized the spatial heterogeneity of the DSR based on the new GeoNEX DSR product. It was shown that mountainous and high-latitude areas are more susceptible to high spatial-temporal variation. DSR products with a resolution of approximately 100 km are insufficient for daily and monthly analyses. Analysis at an hourly temporal scale requires DSR data with spatial resolutions finer than 10 km.

**Author Contribution**

RL: Data curation, Formal analysis, Investigation, Methodology, Writing - original draft, Writing - review & editing. DW: Conceptualization, Funding acquisition, Supervision, Data curation, Formal analysis, Methodology, Writing - review & editing. WW and RW: Funding acquisition, Resources, Writing - review & editing.

**Competing interests**

The contact author has declared that neither they nor their co-authors have any competing interests.

**Acknowledgements**

We would like to thank the AMERIFLUX, Baseline Surface Radiation Network (BSRN), Surface Radiation Network (SURFRAD), and FLUXNET for maintaining and providing their measurement data. We would also like to acknowledge the NASA GeoNEX Earth Exchange data portal for providing the gridded AHI and ABI TOA reflectance data, NASA MODIS Land Science Team for providing MCD43 data, NASA Goddard Earth Sciences Data and Information Services Center for providing MERRA-2 data, and Earth Resources Observation And Science Center for providing the GTOPO30 data. The surface albedo climatology data are downloaded from http://glass.umd.edu/albedo_clim/albedo_climatology_0.05CMG/.

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
