# Peer review of "A GeoNEX-based high spatiotemporal resolution product of land surface downward shortwave radiation and photosynthetically active radiation"

_Earth System Science Data, 2022_

## Author Comment (AC2)

New-generation geostationary satellites have provided us with more chances to investigate diurnal to seasonal vegetation dynamics. Recently, more and more studies have widely used high temporal resolution geostationary satellite datasets and solar radiation data is one of the most required products in many topics. Although this study has no strong novelty for generating DSR and PAR Although this study has no strong novelty for generating DSR and PAR, providing a high temporal resolution product itself has a strong advantage.

Major comments

The author highlight "GeoNEX ...". Considering the unique geostationary satellite network, the strong advantage should be the larger spatial scale integrating the full disk of each geostationary satellite. For example, even if there is no significant improvement in the performance, providing global hourly DSR/PAR itself would have great importance. In this context, this study needs to include Meteosat which covers Europe and Africa. As the authors know, several studies already reported continental-scale hourly solar radiation products using a single geostationary satellite, not GeoNEX data. If the focus of this study is limited to generating continent-scale geostationary satellite-based radiation products, further novelty of the study is required.

*We thank the reviewer for the valuable suggestions! GeoNEX is a collaborative project to provide the enhanced data access to new generation geostationary satellites across the world. The current archive includes imagery data from Himawari, GOES-East and GOES-West. The recent launch of the first Meteosat Third Generation (MTG) satellite will enable us to expand the spatial coverage to Europe and Africa in the future.*

*As the reviewer pointed out, several data sets of DSR and/or PAR from ABI or AHI have already existed. Through literature survey, we have found at least six geostationary DSR/PAR products produced by various institutes. However, the existing products are usually produced from a single source of satellite data. Some data sets are not operational and do not provide convenient data access. In terms of data accuracy, there is also space for improvement. The new GeoNEX DSR/PAR product is an operational gridded high spatiotemporal resolution product with improved accuracy derived from multiple satellites. It has the following strengths:*

1. *The new product has higher accuracy than other existing products.*
2. *The new product is gridded and structured by tiles (600 by 600km), convenient for data transfer and analysis.*
3. *The GeoNEX data have gone through strict geometric correction to remove residual georegistration errors and terrain effects.*
4. *A consistent data product is provided across various satellite sensors.*
5. *It is an operational product and will provide continuous coverage.*
6. *The data are made publicly available through the NASA GeoNEX data portal. It is freely accessible to all the users and no registration is needed.*

Minor comments

Line 78-80: Too vague expression.

*The sentence will be rephrased in the revision.*

Line 82-83: To highlight this point, the author should consider global coverage.

*Thanks for the suggestion! As discussed in the response to the general comment, this is exactly our goal. More details will be provided in the revised text.*

Line 82-85: Just using one more geostationary satellite cannot be the novelty of this topic.

*As elaborated in the response to the general comment, we will provide additional justification here.*

Line 85-86: Out of context.

*It will be removed from the revision.*

Line 87-89: How LUT method address the research gap which the author mentioned in the above paragraph?

*We will rephrase this paragraph in the revised text. The justification to use the physical LUT-based approach is to provide the highly accurate DSR/PAR retrieval when the products of atmospheric parameters from these geostationary data are limited.*

Table1: The author highlights the higher spatial resolution (1km) of this study, but input TPW from MERRA2 has over 50km spatial resolution. Is it acceptable?

*This is a great point! The coarse resolution TPW data were used in the current version, given the relatively low spatial variability of TPW. We will consider directly predicting TPW from the geostationary data and evaluate its impact on the DSR and PAR estimation in the future.*

Section 3.1.3 is well examined the uncertainty of large VZA, which is critical in geostationary satellites. Section 4.1 also well highlighted the advantage of the geostationary satellite-based product.

*Thanks for the encouraging comments!*

---

## Author Response (AR1)

Dear Dr. Hao and Anonymous Reviewers:

We would first like to thank you all for editing and reviewing our manuscript entitled "*A GeoNEX-based high spatiotemporal resolution product of land surface downward shortwave radiation and photosynthetically active radiation*". Your valuable comments and suggestions are highly appreciated, which greatly improved the quality of our study.

Following the comments, we have carefully revised the manuscript. We paid special attention to highlight the novelty and contributions of the study, providing clarification on the strength and uniqueness of the presented data set. Compared to existing products, the new GeoNEX DSR and PAR product is an **operational, easy-to-access, convenient-to-use, gridded product with unparalleled accuracy from multiple satellites**, which is expected to eventually provide continuous global coverage between 60°N and 60°S. We also added new results of sensitivity analysis, provided additional information on the algorithm and technical details of the data set, and made other changes as suggested.

The high-quality fine spatiotemporal resolution product of DSR and PAR is in great need by users from many disciplines. Actually, we have received a number of inquiries about the new product from users around the world just after they read this under-review manuscript. We sincerely hope this high-quality operational DSR and PAR product with substantially improved accuracy can be used to address various scientific and application questions.

Together with the letter, we enclose a point-by-point response to all the review comments.

Thank you again for your time and efforts!

Sincerely,

Dongdong Wang on behalf of the co-authors
Associate Professor
Department of Geographical Sciences
University of Maryland, College Park, MD 20742, USA
ddwang@umd.edu

===============Comments from Reviewer 1===============

This paper presents a LUT-based method to generate Surface downward shortwave radiation (DSR) and photosynthetically active radiation (PAR) products with ABI and AHI measurements. The LUT-based method used in this paper, was initially developed by Liang (2006), and then extended by Wang et al. (2020) for MODIS DSR/PAR product (MCD18). Actually, this LUT-based method was widely used many times, such as Zhang et al., (2014, 2019), and many other authors. Thus, highlights of this paper are not significant. Too much repeatability work. In addition, direct and diffuse components of DSR or PAR are not calculated by this study, only having global DSR or PAR. I suggest to refine your highlights and avoid to do repeat work.

Liang, S., Cheng, J., Jia, K., Jiang, B., Liu, Q., Xiao, Z., Yao, Y., Yuan, W., Zhang, X., Zhao, X., Zhou, J., 2021. The Global Land Surface Satellite (GLASS) Product Suite. Bulletin of the American Meteorological Society 102, E323–E337.

Wang, D., Liang, S., Zhang, Y., Gao, X., Brown, M.G.L., Jia, A., 2020. A New Set of MODIS Land Products (MCD18): Downward Shortwave Radiation and Photosynthetically Active Radiation. Remote Sensing 12.

Zhang, X., Liang, S., Zhou, G., Wu, H., Zhao, X., 2014. Generating Global LAnd Surface Satellite incident shortwave radiation and photosynthetically active radiation products from multiple satellite data. Remote Sensing of Environment 152, 318–332.

Zhang, X., Zhao, X., Li, W., Liang, S., Wang, D., Liu, Q., Yao, Y., Jia, K., He, T., Jiang, B., Wei, Y., Ma, H., 2019. An Operational Approach for Generating the Global Land Surface Downward Shortwave Radiation Product From MODIS Data. IEEE Trans. Geosci. Remote Sensing 57, 4636–4650.

*We thank the reviewer for the valuable comments! We are sorry about the confusion the current manuscript had left. We agree with the reviewer that the contribution of this manuscript is not in the area of algorithm development. Instead, as a data description paper, this study presented the new operational GeoNEX DSR and PAR product with unparalleled quality.*

*The main highlight of the study is the high accuracy of the data product itself. It presents about 15% and 20% improvement compared with the commonly-used CERES product at hourly and daily scale respectively. Although several geostationary DSR/PAR products have already been generated, the existing products are typically based on a single satellite data source and available in the satellite map projection. Some data sets are not operational and do not provide convenient data access. In terms of data accuracy, there is also space for improvement. The other innovation of the study is applying a mature algorithms over a distinct data source, the enhanced collection of multiple new-generation geostationary satellite data archived through the GeoNEX platform. The GeoNEX archive is not simply another copy of the collected geostationary satellite data. It removed the residual geometric errors, applied the orthorectification correction and provided the pixel-level accurate view geometry information. Besides, the data from various satellite sensors are stored in a consistent global tile gridding system.*

*The retrieval algorithm was originally used to product the NASA operational MODIS DSR and PAR product (MCD18). It is the first time that the LUT-based retrieval approach has been applied to the new generation geostationary data. Although the GeoNEX DSR and PAR adapted the heritage algorithm of the MCD18, its accuracy is much higher than that of MCD18 because of the high temporal resolution of geostationary data.*

*Regarding the question on the partition of diffuse and direct radiation, this LUT-based approach is able to produce these components. However, we did not see the superior performance of the diffuse partition as the global radiation has. As a result, this version does not include the diffuse radiation. We will continue working on the algorithm improvement and add the components in the future data release.*

*We thank the reviewer again for the suggestions. We have refined the introduction and conclusion part to further highlight the advantages of this product.*

===============Comments from Reviewer 2===============

New-generation geostationary satellites have provided us with more chances to investigate diurnal to seasonal vegetation dynamics. Recently, more and more studies have widely used high temporal

resolution geostationary satellite datasets and solar radiation data is one of the most required products in many topics. Although this study has no strong novelty for generating DSR and PAR Although this study has no strong novelty for generating DSR and PAR, providing a high temporal resolution product itself has a strong advantage.

Major comments

The author highlight "GeoNEX ...". Considering the unique geostationary satellite network, the strong advantage should be the larger spatial scale integrating the full disk of each geostationary satellite. For example, even if there is no significant improvement in the performance, providing global hourly DSR/PAR itself would have great importance. In this context, this study needs to include Meteosat which covers Europe and Africa. As the authors know, several studies already reported continental-scale hourly solar radiation products using a single geostationary satellite, not GeoNEX data. If the focus of this study is limited to generating continent-scale geostationary satellite-based radiation products, further novelty of the study is required.

*We thank the reviewer for the valuable suggestions! GeoNEX is a collaborative project to provide the enhanced data access to new generation geostationary satellites across the world. The current archive includes imagery data from Himawari, GOES-East and GOES-West. The recent launch of the first Meteosat Third Generation (MTG) satellite will enable us to expand the spatial coverage to Europe and Africa in the future.*

*As the reviewer pointed out, several data sets of DSR and/or PAR from ABI or AHI have already existed. Through literature survey, we have found several geostationary DSR/PAR products produced by various institutes. However, the existing products are usually produced from a single source of satellite data. Some data sets are not operational and do not provide convenient data access. In terms of data accuracy, there is also space for improvement. The new GeoNEX DSR/PAR product is an operational gridded high spatiotemporal resolution product with improved accuracy derived from multiple satellites. It has the following strengths:*

1. *The new product has higher accuracy than other existing products.*
2. *The new product is gridded and structured by tiles (600 by 600km), convenient for data transfer and analysis.*
3. *The GeoNEX data have gone through strict geometric correction to remove residual georegistration errors and terrain effects.*
4. *A consistent data product is provided across various satellite sensors.*
5. *It is an operational product and provides continuous coverage.*
6. *The data are made publicly available through the NASA GeoNEX data portal. It is freely accessible to all the users and no registration is needed.*

Minor comments

Line 78-80: Too vague expression.

*The sentence has been rephrased. Please see Line 66-69.*

Line 82-83: To highlight this point, the author should consider global coverage.

*Thanks for the suggestion! As discussed in the response to the general comment, this is exactly our goal. Additional description on this was added. Please see Line 406-409.*

Line 82-85: Just using one more geostationary satellite cannot be the novelty of this topic.

*As elaborated in the response to the general comment, we have clarified the novelty and strength of the new product. Please see the additional statements in the introduction (Line 76-86) and conclusion (Line 401-409) sections.*

Line 85-86: Out of context.

*It has been removed from the revision.*

Line 87-89: How LUT method address the research gap which the author mentioned in the above paragraph?

*We have rewritten this entire paragraph and restructured the justification of the new product. Please see Line 76-86.*

Table1: The author highlights the higher spatial resolution (1km) of this study, but input TPW from MERRA2 has over 50km spatial resolution. Is it acceptable?

*This is a great point! The coarse resolution TPW data were used in the current version, given the relatively low spatial variability of TPW. We will consider directly predicting TPW from the geostationary data and evaluate its impact on the DSR and PAR estimation in the future.*

Section 3.1.3 is well examined the uncertainty of large VZA, which is critical in geostationary satellites. Section 4.1 also well highlighted the advantage of the geostationary satellite-based product.

*Thanks for the encouraging comments!*

================Comments from Reviewer 3================

This study presented a new DSR and PAR dataset derived from GOES-R and Himawari at high spatial (1 km) and temporal (hourly) resolutions. The dataset achieved <20% and <10% relative errors for hourly and daily DSR, respectively, which were claimed to be higher than existing datasets. The manuscripts demonstrated the benefits of high spatial and temporal resolutions, and therefore partly justified the importance of developping this new dataset. In particular, Figure 9 is interesting, revealing that high resolution is critical for hourly radiation. However, I'm not convinced by this study for the following reasons:

1. The innovation is questionable. There are already many radiation datasets derived from geostationary satellite data, either from GOES-R or Himawari. Some of them are also high resolution. The manuscrit needs to clearly address the questions: why do we need a new one? What's the advantage of this study, e.g., distinct data sources or distinct algorithm?

*We thank the reviewer for the valuable comments and suggestions! One major contribution of the study is the high accuracy of the new operational data product, with significant improvement comparing with the existed products. Through literature survey, we have identified several other products of DSR or PAR from various geostationary data. These products are typically based on a single satellite data source and available in the satellite map projection. Some data sets are not operational and do not provide*

*convenient data access. To address these issues, we adapted a mature physics-based retrieval algorithm to the enhanced collection of multiple new-generation geostationary satellite data archived through the GeoNEX platform to produce a new operational high spatiotemporal resolution product of DSR and PAR with improved data accuracy.*

*This physics-based retrieval algorithm has been initially developed for the operational NASA MODIS DSR and PAR product (MCD18). The extensive quality assessment of MCD18 showed that this algorithm is reliable, efficient, and highly accurate. It is the first time to adapt this algorithm to the new-generation geostationary satellite data. The product validation and comparison has demonstrated the superior performance of this algorithm over the geostationary data compared to other alternatives.*

*The GeoNEX project provides the enhanced access to multiple geostationary satellite data across the world. The GeoNEX archive is not simply another copy of the collected geostationary satellite data. It removed the residual geometric errors, applied the orthorectification correction and provided the pixel-level accurate view geometry information. Besides, the data from various satellite sensors are stored in a consistent global tile gridding system. All these preprocessing steps created the foundation for producing the high quality DSR and PAR product.*

*To be specific, the GeoNEX DSR/PAR product has the following features:*

1. *The new product has higher accuracy than other existing products.*
2. *The new product is gridded and organized by tiles (600 by 600km), convenient for data transfer and analysis.*
3. *The GeoNEX data have gone through strict geometric correction to remove residual georegistration errors and terrain effects.*
4. *A consistent data product is provided across various satellite sensors.*
5. *It is an operational product and will provide continuous coverage.*
6. *The data are made publicly available through the NASA GeoNEX data portal. It is freely accessible to all the users and no registration is needed.*

2. As a data paper, the Method part is too short. A flow chart is needed, including graphyical links between Eq. (1), Eq. (2), inputs and outputs.

*It is a great suggestion! A flow chart and more details have been added on the methodology description. Please see Figure 1 and Line 111-124.*

3. Terrain effect was not considered. Considering many mountain areas are involved, this could be a big limitation.

*Thanks for this valuable comment! The terrain effect was partially taken into account in the current study through two ways. 1) The GeoNEX collection has applied the orthorectfication correction to mitigate the topographic relief effect. 2) The retrieval algorithm has used the altitude-dependent LUT files to handle the impacts of elevation on DSR and PAR. Meanwhile, we acknowledged that the impacts of aspect and slope were not considered and need to be addressed in the future study.*

4. As a data product, no detailed QC and quantitative uncertainty was provided. This is also a big limitation.

*We are sorry for the confusion on QC. Actually, we did include a layer of QC to the product files to indicate the quality of each pixel. We added more description on this in the revision. Please see Line 398-399.*

5. The sensitivity to inputs/parameters could provide deeper insights for potential users.

*Thanks for the suggestion! We added a new section on sensitivity study. Please see Line 282-297.*

6. There was no map of the DSR and PAR products in the manuscript.

*Maps of DSR and PAR was added. Please see Figure 3.*

7. Temporal coverage of the dataset was not mentioned. Is it operational and real time?

*It is an operational product. We added the information. Please see Line 401-402.*

8. Why does this dataset has higher accuracy than other geostationary-based dataset? If high resolution only matters for hourly data, why does this dataset has much lower errors than others at daily scale?

*The high quality of the new GeoNEX DSR and PAR product can be attributed to two major reasons.*

*1) The retrieval algorithm is a highly mature one that has been continuously improved and refined through decade long efforts. Our recent intercomparison study (Wang et al, 2021, RSE) has demonstrated the reliability and accuracy of this physics-based algorithm, especially when no reliable atmospheric products are available.*

*2) The enhanced geostationary data from GeoNEX were used as input. The GeoNEX data processing steps include removing residual geometric errors, applying orthorectification and calculating the pixel-level accurate view geometry information.*

*Because of the high accuracy of the hourly GeoNEX data, the daily data, which were aggregated from the hourly values, also show superior performance. The high quality of the GeoNEX daily DSR/PAR data are not only the result of high spatial resolution. Actually as shown in the manuscript, the data at coarser temporal resolutions have less dependency on spatial resolutions.*

*Reference:*

*Wang, D., Liang, S., Li, R., & Jia, A. (2021). A synergic study on estimating surface downward shortwave radiation from satellite data. Remote Sensing of Environment, 264, doi:10.1016/j.rse.2021.112639*

---

## Author Response (AR2)

The reviewers are satisfied with the authors' revision. I have few comments for the authors' consideration:

*Response: We appreciate the comments and suggestions of the editor and the reviewers.*

1. Avoid using the red-green pair (which is not blind-friendly) in the figures.

*Response: We checked all the figures and re-drew all the affected figures to assure all the plots accessible to those with color vision deficiencies.*

2. The captions in some figures (e.g., Fig. 8) are oversimplified. Please add more details to make then clearer.

*Response: We have gone through all the figure captions and revised accordingly to make them clear and informative.*

3. Fig. 8, please adjust the blank space between the top and bottom panels. Also check other figures' format.

*Response: We updated Figure 8 by reducing the gap between the panels. We also checked all the other figures.*